# *"You could find a good or a bad provider, I would say you just have to go and see"*: A qualitative study of the influence of perceptions of service quality on family planning service use in Burkina Faso

Sarah Castle[1], Fiacre Bazie[2], Amelia Maytan-Joneydi[3], Kindo Boukary[2], Lisa M. Calhoun[3], Yentema Onadja[2], Georges Guiella[2], Ilene S. Speizer[2,4]*

1 Independent Consultant, London, United Kingdom, 2 Institut Supérieur des Sciences de la Population (ISSP) at the Université Joseph Ki-Zerbo, Ouagadougou, Burkina Faso, 3 Carolina Population Center, The University of North Carolina at Chapel Hill, Chapel Hill, NC, United States of America, 4 Department of Maternal and Child Health, Gillings School of Global Public Health, University of North Carolina at Chapel Hill, Chapel Hill, NC, United States of America

* ilene_speizer@unc.edu

**Data Availability Statement:** Data for this study were collected from Focus Group Discussions.

## Abstract

This qualitative study from Burkina Faso explores community-level perceptions of family planning (FP) service quality among FP users and non-users. It examines how perspectives on service quality may influence women's motivation to seek modern methods from health facilities. For this study, twenty focus group discussions were undertaken with non-users and current users of modern FP including unmarried, sexually active women ages 15–19 and 20–24 and ever married women ages 15–24 and 25+ in Bobo Dioulasso and Banfora, Burkina Faso. The findings demonstrate that respondents prioritized a welcoming environment, positive provider-client exchanges, the full provision of information (especially about side-effects), a pain-free experience, a short waiting time, and privacy and confidentiality. Poor service quality did not, in general, reduce women's demand or need to use a FP method. Some women who were reluctant to use formal health services used a non-facility-based method (calendar method, withdrawal, condoms or abstinence). Importantly, many unmarried, younger women and adolescents, who were more likely to be stigmatized by providers, exhibited agency by proactively seeking a method despite the potential for a negative experience. They prioritized their health and wellbeing over and above any interpersonal barriers they were likely to encounter. Incorporating strategies to improve the quality of FP services based on locally defined elements of quality should be a specific programmatic goal. These strategies can be identified through quality assessments employing a woman-centered lens. Women who visit facilities can be encouraged to share their positive experiences with their networks to improve community-level perspectives of facility quality. Improving service quality can attract new users, especially adolescents, and retain those who have already adopted a FP method. Through these multi-pronged actions, women's

Focus Group Discussion Guides and information on the data can be found at the following site: https://dataverse.unc.edu/dataverse/fafcburkinafaso.

**Funding:** This work was supported, in whole or in part, by the Bill & Melinda Gates Foundation [INV-009814 to ISS]. Under the grant conditions of the Foundation, a Creative Commons Attribution 4.0 Generic License has already been assigned to the Author Accepted Manuscript version that might arise from this submission. The authors also received general support from the Population Research Infrastructure Program through an award to the Carolina Population Center (CPC) (P2C HD050924) at the University of North Carolina at Chapel Hill. The contents of this article are solely the responsibility of the authors and do not necessarily represent the official views of CPC or the Bill & Melinda Gates Foundation. The funders had no role in study design, data collection and analysis, decision to publish, or preparation of the manuscript.

**Competing interests:** The authors have declared that no competing interests exist.

(and community) expectations and experience of quality can improve. This, in turn, may lead to greater client satisfaction and associated higher FP prevalence.

## Introduction

Universal access to high quality sexual and reproductive health services is a global goal [1]. When service quality is poor, this may discourage family planning (FP) adoption and continuation and possibly lead to unintended pregnancies among women who are not willing or able to visit a facility [2–6]. To date, few studies have documented the acquisition and transmission of community-level perceptions of service quality and the influence of these perceptions on service use behaviors.

FP use is low in Burkina Faso, the site of this study, and reflects similar trends throughout most of francophone West Africa [7]. In 2021, estimates for modern FP prevalence among married women in Burkina Faso placed it at 31.9%. While this is higher than other francophone West African settings (ranging from 14.2% in Benin to 27.5% in Senegal (From FP2030 Data Dashboard based on estimates from 2021.)), it is lower than other countries in sub-Saharan Africa. Furthermore, about a quarter of married women (23%) in Burkina Faso are estimated to have an unmet need for family planning [7].

Reasons for low uptake of modern FP methods in Burkina Faso and elsewhere in francophone West Africa have been widely documented and include both demand- and supply-side barriers to use [8]. On the demand-side, widespread male disapproval of FP (leading to many women using a method covertly); opposition by community and religious leaders; social norms encouraging early childbearing, and high fertility linked to women's status have been found to be related to non-use [8–10]. In addition, a considerable barrier to use throughout sub-Saharan Africa comprises myths and rumors about FP methods [11, 12]. These include the notion that the use of hormonal methods, in particular, will lead to sterility [12, 13]. This can diminish the desire to use a method, even among those women who report wanting to delay or avoid childbearing as well as among men [8, 14].

On the supply-side, an important factor influencing non-use is the poor quality of FP services [15, 16]. Globally, and in the broader health field, it has been documented that *perceptions* of service quality, and not just *actual* service quality, are important drivers of health service use [17, 18]. Correspondingly, studies of FP programs have demonstrated that inferior service quality results in low levels of attendance and client satisfaction and higher rates of contraceptive discontinuation [5, 6].

Typically, the quality of FP services is defined in relation to services and commodities available at a facility as well as in relation to perceived provider competence, the existence of follow-up mechanisms (for example for LARC removal), and client-provider relations [19, 20]. For young and unmarried women in Burkina Faso and elsewhere, provider interactions have been documented as being particularly key to determining client appraisals of service quality and the intention to revisit facilities [21–23]. The fear of stigmatization of young unmarried sexually active women by providers in cultural contexts where premarital relations are taboo, and childbearing is valued, has been widely documented as leading to the non-use of services [23, 24] and confirmed by the new evidence presented here.

This study contributes to the earlier literature by examining recently collected qualitative data from Burkina Faso that sought to understand how perceptions of service quality can influence both FP uptake and continued contraceptive use and how perceived quality may differentially affect service use of younger and older women.

## Methodology

This study uses qualitative data from focus group discussions (FGD) with women from two cities in Burkina Faso to identify perceptions of service quality (positive and negative) and to examine how these, together with women's own or others' experiences, affect service use. The discussions also identified mitigation strategies used by some women to obtain FP methods despite their own or others' accounts of poor perceived service quality. Study findings inform recommendations for strategies to improve client-centered service quality as well as to support women's agency and motivation to use FP, regardless of their perceived or actual experiences with care.

This qualitative study was undertaken by the Carolina Population Center at the University of North Carolina at Chapel Hill, USA in collaboration with the Institut Supérieur des Sciences de la Population (ISSP) at the Université Joseph Ki-Zerbo, Ouagadougou, Burkina Faso. It took place in November 2021 in two urban/peri-urban areas of Burkina Faso: Banfora and Bobo-Dioulasso. These sites were selected as they corresponded to locations where Pathfinder International was implementing the Beyond Bias program (https://www.pathfinder.org/projects/beyond-bias/). As part of the Beyond Bias program, providers within selected facilities were trained on provision of services to all young people, no matter their age, marital status, or parity. This study recruited participants from communities near facilities where the Beyond Bias project worked as well as near facilities where the Beyond project did not work to determine if community perceptions of quality differed by program implementation site. Since no differences were observed across intervention and comparison sites, these distinctions are not discussed in the findings below. Table 1 shows that the study comprised 20 FGD with groups of about 8–10 women.

Prior to beginning data collection for this study, the ISSP study team met with city and community leaders to explain the study and obtain their approval and support to work in the study settings. Three sites, each in the vicinity of a health facility, were identified in each city to get a mix of urban and peri-urban sites. Once introductory meetings were completed, the

**Table 1. Number of FGDs by type of participant, Burkina Faso, 2021.**

| Age group | Marital status | FP user status | Number of FGDs |
|---|---|---|---|
| **City : Banfora** | | | |
| 15–19 | Never married | Non-user | 2 |
|  | Never married | Current user | 2 |
| 20–24 | Never married | Non-user | 1 |
|  | Never married | Current user | 1 |
| 15–24 | Married | Non-user | 1 |
|  | Married | Current user | 1 |
| 25+ | Married | Non-user | 1 |
|  | Married | Current user | 1 |
| **City : Bobo-Dioulasso** | | | |
| 15–19 | Never married | Non-user | 2 |
|  | Never married | Current user | 2 |
| 20–24 | Never married | Non-user | 1 |
|  | Never married | Current user | 1 |
| 15–24 | Married | Non-user | 1 |
|  | Married | Current user | 1 |
| 25+ | Married | Non-user | 1 |
|  | Married | Current user | 1 |

study team engaged with community leaders in each site; these people were selected to be women and men who run community organizations or associations and were knowledgeable about the realities and needs of people in their communities. These people were asked to help identify eligible women in their communities for the focus group discussions based on age, marital status, and sex. We avoided mentioning the specific subject of the interviews to avoid biasing who was proposed as participants. It was made clear to the leaders that we were asking them to propose potential participants but that the participants would have the right to accept or refuse participation. Thus, on a specific day, community leaders were asked to recruit about 20 unmarried women ages 15–19 (or unmarried 20–24, or married 15–24, or married 25 + years) and from this pool of eligible respondents, interviewers determined what type of focus group to undertake. On the day of the interview, the eligible participants (by age/marital status group) were screened by the project team to determine if they were current non-users or current users of a modern method of contraception including pills, injectables, implants, and the IUD; those who were only using condoms were considered non-users of a facility-based method and were grouped with the "current non-users" for a FGD. This permitted determining from the group of women, which FGD(s) could be done that day (i.e., if participants were mostly users, they created a user group or alternatively, if participants were mostly non-users they created a non-user FGD group). Any woman who was not eligible for the FGD on the day of interview was either given an appointment for another day or thanked for her willingness to participate and given a refreshment.

Each eligible woman was read the consent form and provided signed consent prior to engaging in the FGD. In total, the team undertook eight FGD with unmarried, sexually active women ages 15–19; four with current non-users and four with current users of a modern method (see Table 1). Four groups were undertaken with unmarried, sexually active women ages 20–24; half were non-users and half were current users. It should be noted that current non-users may have included women who had used a method in the past. The remaining 8 groups were among ever married women ages 15–24 (four groups) and ever married women ages 25+ (four groups) stratified by their current use experience. Notably, given that women who are married are likely to have a similar experience whether they are 15–19 or 20–24, we grouped the married women in these two age groups together.

A semi-structured interview guide developed in French was employed; the guide covered general information on sexual and reproductive health seeking behaviors, factors that the community like about FP services, factors that the community dislike about FP services, where women (and young people) learn about FP service quality, and the influence of perceived quality on FP and facility use behaviors. The guide was translated into local languages and pilot tested during training of the study team and subsequently revised for clarity of terms and approach.

Each FGD was led by a trained moderator and notetaker and conducted in Dioula or Mooré languages. Each FGD lasted about 90 minutes and was recorded and transcribed into French. The transcriptions were uploaded into Dedoose, a web-based qualitative analysis software that allows for collaboration [25]. Transcriptions were coded in French by four coders (AMJ, FB, KB, and SC). An initial code book was developed through review of the focus group guide and of the pilot data. All coders then jointly coded two transcripts until consensus was reached over code application. Agreement over coding was achieved through joint review and discussion of all the codes applied by each coder in the jointly coded transcripts. Following discussion of the joint coding of the two transcripts, modifications and a number of new emergent codes were implemented to the code book. Once consensus was reached, coders then individually coded the remaining transcripts using the finalized code book with a second coder reviewing the coding by the first coder. Thematic analyses were undertaken to create

matrices to identify themes, connections, and patterns around how women conceptualized various elements of service quality and their respective influence on service use. Where relevant, distinctions in responses by the age, marital status, or use status of the respondent were identified and discussed. The quotes presented in this paper were translated from French into English by the first author and reviewed and approved by the Burkina Faso study team to ensure accurate interpretations.

### Ethics declarations

All study materials including consent forms, focus group discussion guides, and the study protocol were reviewed and approved by the ethics committee in Burkina Faso (Comite d'Ethique pour la Recherche en Sante #2021–000133) and at the University of North Carolina at Chapel Hill (#21–1898). All participants provided signed consent to participate, this included the participants who were ages 15–17. Given that participants were recruited in community settings, it was not possible to obtain parental consent for participation; this strategy was approved by the two ethics committees listed above.

## Results

This study explores positive and negative perceptions of service quality based upon the testimonies generated during the focus group discussions. In some cases, information provided was obtained from others in the respondents' networks whereas, in other cases, women recounted their own lived experiences. The study intentionally recruited women at the community-level and not via facilities. This allowed for a range of perspectives around service quality from experienced users (but not immediately following a clinic visit) as well as from non-users who may have different views relating to quality that may affect their future motivation to use a method. Below we summarize the results in three sections: the impact of positive perceptions of quality on facility use; the impact of negative perceptions of quality on facility use; and mitigation strategies for women potentially discouraged from visiting a facility by their experiences or the testimonies of others. These mitigation strategies are often driven by personal agency to seek a service.

### The impact of positive perceptions of service quality

Regardless of age or marital status, both current users and current non-users said that positive perceptions of service quality would influence their use of FP services.

*It is as if the person has marked out the path so that you can go as well–so I will go to see for myself. I will say to myself "well the other person was treated well and so I will also be treated well there."And in turn, I will tell other people.*

Banfora, unmarried users ages 20–24

*According to me the good experiences that you heard can make you happy and you will decide to go to a family planning center. If someone tells you of good experiences, that will encourage you to go to FP services!*

Banfora, married non-users ages 15–24

The most influential component of women's positive perspectives largely pertained to the 'welcome' given to clients by the providers which was the main concern of younger women interviewed. In particular, unmarried women shared information about services where the

providers were friendly and kind (despite clear evidence of their pre-marital sexual activity which is socially taboo) and, because of this, they encouraged others to go.

> *If I go to a health center and I am well treated, it will influence me–I will go and keep going. Even if someone comes to ask me, I will tell them to go there. I will even tell my friends to go to this facility because they do their work well.*

> Bobo Dioulasso, unmarried users ages 20–24

The warm welcome motivated those who had already visited a facility to return and encouraged those who had never used a FP method to seek services.

> *A married woman already went (to a facility) and came back and shared information about her positive experience with me- and then I wanted to go! If she didn't like it, I would not go— if she comes back and says she was treated nicely and the providers chatted with her, that can change your point of view. It depends on the staff–if she was treated nicely, she will share her experiences with other people. When the service is good, everyone will go–even me!*

> Bobo Dioulasso, unmarried users ages 20–24

> *For those who have not yet used family planning, if they hear about the positive experience we have had there that encourages them to go. I will tell them that they (the providers) are not going to shout or yell at you.*

> Banfora, unmarried users ages 15–19

Those in the youngest age groups were generally concerned about side-effects or the potential pain they perceived to be associated with LARC insertions. Discussions with users they knew meant that they learned about pain-free experiences at the facility and these helped to allay their fears.

> *I was at home and an auntie told when she went to get family planning it didn't hurt and that's what motivated me to go myself.*

> Bobo Dioulasso, unmarried users ages 15–19

Young women also revealed that they share their positive experiences about lack of side-effects and reassured their friends which helped to catalyze the latter's service use.

> *It's possible, if I go and they nicely treated me, my periods will come as normal–and it won't bother me. If you tell this to your best friend, then she can go too. If what you tell her is true and she too does not have problems, then she can go and tell other people.*

> Bobo Dioulasso, married users ages 15–24

> *(Hearing positive experiences) can have a big influence because the person comes to you to chat and influence you–what people talk about rumors related to family planning–some say it's good and some say it's bad. It gives you a better idea about it and you can go and listen to what they (the providers) have to say without fear.*

> Bobo Dioulasso, married non-users ages 15–24

The reports of positive experiences at the facilities were also said to create a feeling of trust between young people and the providers if they did eventually decide to use services.

*I think that it (a report of a positive experience) would create trust between the adolescents and the providers. It could be a chance for them to talk to their friends about family planning and encourage them to go and get it.*

Banfora, married users ages 25+

Perceptions of good service quality and shared experiences also influence older women who seek to space their births or stop childbearing altogether as well as younger women who wish to delay their first pregnancy.

*It is possible that there are women who have lots of children but in spite of this they wish to stop. If you say to such a person "ah if you go there you will get good advice about spacing your children"(they will listen). There are some women who have lots of children and they can't even feed them. If you go to them and counsel them and you say that the provider speaks nicely, that can influence the person to stop getting pregnant or to space her births.*

Bobo Dioulasso, unmarried users ages 15–19

Testimonies from the FGDs also indicated that, once convinced of good service quality, women would repeatedly visit the same facility. This may be important in terms of reducing discontinuation and increasing long-term FP use.

*If I go there and I have been well treated, next time I would not hesitate to go again because I would say to myself that they treated me well. I wouldn't even think twice! I'd go there directly!*

Bobo Dioulasso, unmarried non-users ages 20–24

Importantly, the sharing of positive experiences could also influence women's husbands to look more favorably upon FP and to support their wives' contraceptive use.

*The positive experiences that married women hear about family planning could encourage those who were hesitating to use it to go to services. In addition, if their husbands hear about others' positive experiences, this will encourage them to give their wives permission to use contraception.*

Banfora, unmarried users ages 15–19

Thus, the sharing of positive experiences appears to have a considerable impact on women's motivation to use services. In particular, the reassurance of a warm welcome seemed to be the key factor in encouraging others to visit a facility. In addition, allaying fears related to possible pain (for example, associated with LARC insertions) and reassuring young users of a climate of trust between the client and provider was also perceived to be important. The sharing of positive experiences influenced new users, including those who wished to delay pregnancy as well as repeat users who felt encouraged to return to services they perceived to be of high quality. Importantly, the sharing of positive information also potentially influenced husbands, and could lead them to look favorably upon FP, engage in more open couple communication, and support their wives' use.

## The impact of negative perceptions of service quality

Women also reported sharing negative perceptions of FP services and acknowledged that these also had an impact on their own and others' use. In the same way that a positive welcome by providers could motivate women to use services, a poor reception could also demotivate them from visiting a facility. The possibility of a poor welcome seemed to particularly discourage young, unmarried, never-users from attending services possibly because they were more likely to receive judgmental attitudes and disapproval on the part of the providers.

*What I don't like is that if you go to family planning services, the providers shout at people and they don't give them good advice. This makes certain (young) women not want to go back.*

Banfora, unmarried non-users ages 15–19

*This type of (bad) behavior by providers can give a bad impression of the health center because everything happens there and the information gets out by hearsay and then everyone is informed–and it can result in the family planning services not being used.*

Banfora, unmarried non-users ages 20–24

Adolescent non-users may be put off even making a first visit, given that their young age and unmarried status could make them more likely to receive stigmatizing treatment.

*People talk about "respect above everything "–if you go (to the clinic) and you are not respected, the next time you will not go back there and you will get the idea in your head that family planning is a bad thing. But if you go and you are warmly welcomed, you will say that it is a good thing and you can encourage other people to go. If you yourself are not warmly received, you can't encourage anyone else to go can you?*

Bobo Dioulasso, unmarried users ages 15–19

*If a married woman goes to a family planning clinic and she is not well received–if you, as an adolescent, you hear that, you will say to yourself "Well! Even a married woman is not warmly welcomed and it will be worse for me as a child–they could send me away!"*

Banfora, unmarried users ages 15–19

Unmarried adolescents also recounted what they perceived to be poor provider technical competence (or rumors about poor competence). Informants felt that these perspectives put women off using or returning to services or indeed recommending them.

*It wasn't me but I heard this from someone–she said that she went (to the family planning services) she had an IUD fitted but it wasn't properly inserted. When she got home, it fell out–so she didn't go back there. She went to get another one in a clinic in Bobo.*

Bobo Dioulasso, unmarried non-users ages 15–19

*When you go, you can come across a provider who is in a bad mood and he can transfer his negative feelings to you. For example, if you are coming to get Norplant, he can force it in you and it will hurt. So when you go home and someone else asks you, you will say it wasn't a nice experience to get Norplant because it was painful. So, in turn that person won't want to go and get a Norplant at the health center. I don't like that.*

Bobo Dioulasso, unmarried users ages 15–19

In addition, a lack of availability of methods also led to negative perspectives and women may warn others about this aspect of poor quality which could lead to others avoiding family planning facilities.

*The fact that you think that you went and then they tell you that there are no methods available- it can discourage you from going again. Their (the providers') indifference or negligence will discourage you from going.*

Banfora, unmarried users ages 15–19

Women who had already used family planning were also reticent of returning to services where they felt they had been badly treated. A number of such informants were currently not using a method at the time of the study.

*What they did to me, if they did that to 100 women, then no one would go there. You go to family planning services for your health–if you go and have problems, why would you go again if it was not to have more problems?*

Banfora, married non-users ages 25+

One informant said that poor treatment by providers at the beginning of a woman's reproductive career could put her off family planning for life. Perceptions of women's first contact with services could therefore potentially be influential with regard to subsequent decision-making and contraceptive use.

*When you are young and you did it (used FP services) and you have a good experience, you will understand. But when you did not have a good experience at the beginning you won't want to do it anymore and you will be discouraged. Everything depends upon how it was at the outset. If the beginning was good, the end will be good. If the beginning was bad, then there won't be an end.*

Bobo Dioulasso, married users ages 25+

Thus, the barriers, which were the subject of exchanges within women's social networks, largely comprised the negative way in which women perceived they were received by providers during their FP consultations. This appeared to be the most frequently cited negative experience and one that seemed to strongly demotivate others, particularly adolescents and younger women, from visiting facilities. A lack of method availability was also cited as a negative factor which may discourage others from seeking services. Importantly, women also strongly emphasized the impact of negative issues they associated with poor provider technical competence including pain on insertion of a method or lack of supplies and provider indifference.

## Mitigation strategies for those discouraged from using modern family planning services

During the discussions, women were asked what would happen if they or their peers felt discouraged from using FP services because of the negative experiences they had been told about or which they themselves had experienced. Most still wanted to space their births and employed mitigation strategies which differed importantly by age, marital status and, in some cases, educational background. Focus group discussion participants indicated that younger women, who were more likely to be poorly received at health centers (or who expected to be

poorly received), especially if they were unmarried, sought to procure modern contraceptive methods elsewhere, for example, from pharmacies. However, older women who often were using long-acting, hormonal methods, were obliged to return to a health care provider, no matter what their expectations, because they needed the services and the assistance of a trained professional for insertion or removal.

*Young women will go to pharmacies to buy pills for example–because when you buy the pill you don't have to give any explanation. But married women who need a method which needs to be inserted by a provider, are obliged to go to the health center in spite of everything because you cannot insert the method yourself.*

Banfora, unmarried users ages 20–24

Other women across different FGDs said that married couples could also resort to withdrawal or condom use if they were not prepared to go to the health facility to get a modern method of FP. These methods require good couple communication which may not always be possible.

*For me, those who hesitate to go to the health center to get family planning services use a condom to avoid getting pregnant.*

Bobo Dioulasso, unmarried users ages 20–24

Many respondents sought to use the calendar method if they did not want to go to the health facility because of expectations of negative treatment as reported by their friends or family. However, the calculations needed to use such 'natural' methods effectively were sometimes perceived to be too complex for uneducated women.

*Those who are afraid of going to the health center and get a medicine (contraception) calculate the days of their menses. So, when their periods come, certain people say you need to not have sex three days beforehand. After your period has finished, you need to leave a week–or something like that. So those who are bright can make this calculation–they don't go to health centers–they master this technique. But uneducated women like me, we go to the facility.*

Bobo Dioulasso, married users ages 25+

As the calendar method requires good couple communication, some younger, unmarried respondents felt that it was more suitable for those in marital unions. Adolescents, perhaps in less stable relationships, said that, in their situation, it was often easier just to abstain from intercourse altogether.

*Adolescents should just abstain from sex–as they are hesitating and don't want to go to the health center. It's just better if they don't have sex at all.*

Bobo Dioulasso, unmarried users ages 15–19

Others reported that women can turn to a variety of traditional methods (the most commonly cited being a 'tafo'–a thread that is attached around the waist) if they do not want to visit a facility. However, women recognized that the efficacy of traditional methods was often somewhat limited and that users end up going back to the FP facility whether they liked it or not.

*It's like a thread–one of my friends told me you attach it around your waist. But as it's not very effective—users fall pregnant and they end up going back to the family planning services to get a modern method.*

Banfora, married users ages 15–24

In addition, when asked what women who did not want to visit a facility would do, a number of respondents talked of going to visit informal vendors who often sold unregulated pills and products which they claimed could prevent pregnancy.

*There are people who wander around selling medicines- they sit down and sell them at the roadside. They are not doctors–they just sell medicines and some people buy contraception from them.*

Bobo Dioulasso, unmarried non-users ages 20–24

Importantly, in some cases, married women felt obliged to (re)visit the facilities or find another facility to visit despite their or their peers' negative experiences as they felt they had no choice. They acknowledged that they still needed family planning, since they are sexually active and want to delay/avoid childbearing, and thus felt obliged to accept the risk of poor treatment.

*For me, as there are no other family planning services, whatever happens you are going to return there. There are no other services here, so there is no question of a choice.*

Banfora, married users ages 25+

*People don't think the same way. There are women who had a bad experience and they simply won't go back. But there are others who will return because they need the services.*

Banfora, married non-users ages 15–24

Thus, the need for contraceptive use may override perceptions of poor service quality as the desire to prevent pregnancy is prioritized.

## Mitigation strategies among younger, unmarried women

As abstinence, withdrawal, the calendar method and condoms all require good couple communication (which may not be forthcoming within young people's early sexual relationships) some young women thought they had no alternative but to seek a modern method at a facility, whether or not they expect to be treated negatively.

*There are some people who count the days (of their cycle) or use condoms so they don't have to go to the health center—that's what married women do anyway. But some people have irregular cycles and you can't count the days. We young women are forced to go to the clinic (to get a method of contraception) but married woman can count the days or use condoms.*

Banfora, unmarried users ages 15–19

*That's what I said! Even if you have not been well treated you need to go there again. You think of how they hurt you, but you don't have the choice. You will go back there. It's up to you to put up with it!*

Banfora, married non-users ages 15–24

Importantly, a considerable number of young respondents who had heard accounts of others' negative experiences with FP services were not put off from visiting facilities themselves. Instead of fatalistically accepting their lot, they often used proactive strategies and exhibited significant agency to still seek services despite the negative reports circulating within their social networks. For example, adolescents, whose hesitation was sometimes based on the reported judgmental manner of a specific provider, spoke of returning to the same health facility when that provider was not on duty.

*I don't think you should make a big deal out of it as you may not come across the same person.–another person can receive you well and give you the method you want as well as show you all the advantages and disadvantages without any problem.*

Banfora, unmarried users ages 15–19

*If they have heard about negative experiences even though they themselves have never been there, they can go and see for themselves. It could be that when they arrive they get a provider who doesn't do the same things they have been told about–so they can come back and tell everyone "I went and had a good experience–they said nice things and I appreciated it."*

Banfora, unmarried non-users ages 15–19

Younger users noted that the poor welcome or negative provider reaction recounted by others could have just occurred on a particular day when the health worker was in a bad mood. They surmised that if they visited subsequently, the same provider may be more accommodating and friendly.

*I think that when you go to a health center, all the days are not the same. As I said, if a provider is angry, he needs to leave his anger at the door before entering the facility. Maybe one client went and found him angry, but another could go and find him in a good mood!*

Bobo Dioulasso, unmarried users ages 20–24

Due to negative reports from others, some respondents spoke of changing facilities even if it meant travelling to another town which also may result in additional costs. Younger women were particularly aware of the potential negative reaction they may receive from providers which may lead to them going to other health centers to get the family planning commodities they needed.

*If a woman goes to a family planning service and she is not satisfied, she can go to another CSPS (public facility). If it's an unmarried woman she can go to another health center because she was not satisfied with the service she received (in the first one).*

Banfora, unmarried users ages 20–24

*This kind of behavior on the part of providers can give a bad image of the health center because information gets around by word of mouth and everyone is aware–this can mean that the health center is not used. There are certain people who will ask others for money to cover the cost of transport to go to Banfora for their family planning services just so they don't have to face up to certain (providers') behaviors here.*

Banfora, unmarried non-users ages 20–24

Others preferred to use the private sector and thought that the higher cost would bring better quality and, in particular, a higher standard of clinic environment and provider behavior. It

appeared that some respondents perceived that free services were associated with poor quality and lack of accountability.

*It depends on the beginning. If I go today to the CSPS (public facility) which is on the hillside to get family planning and the provider is disrespectful or does something which I don't like, the next time I will go to the clinic. There it is private and you are a 'client' but at the CSPS it is free. When it is free, everyone likes that and so the centers are full. But in the private clinics there is space, a TV and a fan which blows cool air and you are happy. It's great but you have to have money because you will be billed and even the TV and the fan don't come free! People change centers, as at the CSPS you are treated bizarrely and you will say to yourself it's because I don't have to pay. If you have money, you can change and go and get family planning in another (private) center.*

Bobo Dioiulasso, unmarried users ages 20–24

Others felt that as long as they got what they came for–i.e., a contraceptive method–they were prepared to deal with poor service quality and, in particular, an unfriendly welcome. It appeared that often, perceptions of service quality were so low, that women simply managed their expectations and put up with the poor treatment to obtain the FP product they desired.

*Even if you go and you are not well received, you shouldn't be worried- you should stay calm! You know what you came to get so try and stay calm, get the service you require and leave. You could go elsewhere and also find that it is like that and then what would you do?*

Banfora, unmarried users ages 15–19

Other respondents felt that they owed it to themselves to procure a method despite the hurdles they had to overcome. This was the case for women (especially older women) seeking a long-acting hormonal method. That said, younger, unmarried respondents also discussed the need to exercise restraint or tolerance to transcend the potential barriers associated with the way they were often received. At times, they proactively sought out a method to enhance their own wellbeing and did not accept having their FP use dictated by providers' negative attitudes. In this case, the young women are prioritizing their health over and above the potentially poor clinic experience; this characterizes elements of agency of these young women [26, 27]. Those who seek FP services appear to value their individual wellbeing which enables them to overcome external barriers related to the potentially negative experience of the facility visit and provider consultation.

*You need to get this service as it is for a good cause. I think you can keep calm and go to the facility. You can't say to yourself that because of such behavior I will not go to this center as you need what's there. So I think you have to hold your head up high and go because you have this need and (if you don't go) it could come back to haunt you.*

Banfora, unmarried non-users 15–19

*I think that if you go and if you are not well received, you should go back the next day because you may get a provider who is more attentive–as there is not just one person doing the work. But if you refuse to go back because you were poorly welcomed by a provider, it is as if you are doing harm to yourself.*

Banfora, unmarried users ages 20–24

Thus, the mitigating factors comprised using alternative FP methods which did not require facility visits to receive services. These included condoms, the calendar method, abstinence or withdrawal, all of which necessitate a degree of partner communication which may not be easy, especially for young or unmarried women. Others preferred to get methods such as pills from the pharmacy or to purchase methods of dubious efficacy from ambulatory vendors. Many remarked that they would resort to traditional methods if they were not comfortable visiting a facility.

There were, however, a significant number of women who said that despite their own previous unsatisfactory visits or learning about the negative experiences from others, they would persist with visiting the facilities. Some of these women were looking for long-acting and hormonal methods which they recognized that they can only get from a facility. In these cases, the women (young and old) argued that a provider's mood could have improved or s/he may not even be there and they would see someone else. Some sought contraception from private providers based on the premise that paying for services would mean that they were likely to be of a higher quality. Younger respondents, in particular, said that even though they knew that they might be poorly received in a public clinic, they would go anyway as, either they had no choice, or they needed the method so badly they were prepared to put up with the negative treatment. However, importantly, many young women saw getting a method, in the face of potential barriers, as necessary to optimizing their wellbeing and exhibited agency and self-efficacy to overcome any obstacles. As such, they were prepared to put up with the temporary hardship of a potentially negative interaction with a provider and actively prioritized their own health above such external factors beyond their control.

## Discussion

The new qualitative evidence from selected settings in Burkina Faso presented here was collected at the community level and adds an important new dimension to the linkages between perceived quality and service uptake and similarly demonstrates that women's perceptions of the quality of FP services has, in many cases, an impact on their motivation to use the services. Improving FP service quality can attract new users, especially adolescents, and retain those who have already adopted a FP method. Women interviewed during FGD prioritized a welcoming environment, positive provider-client exchanges, the full provision of information (especially about side-effects), a pain-free experience, a short waiting time, and privacy and confidentiality. Improving quality of care with a person-centered focus [28, 29] means addressing these issues that are identified as important to women who are users and non-users of services.

The study found that poor service quality did not, in general, reduce women's demand or need for FP nor their use of methods. Women mitigated their reluctance to use formal health services by employing other strategies to control their fertility such as using the calendar method, withdrawal, condoms, abstinence, or traditional methods–most of which require good partner communication which may not always be feasible. Other women visited pharmacies or ambulatory vendors to get oral contraceptive pills or other methods; this reduced their need to engage with health facility staff. The fact that women employ alternative methods and strategies to avoid a pregnancy means that poor service quality may increase unmet need for modern FP and unintended pregnancy as women seek to control their fertility but are put off using facility-based services by their prior experiences or the testimonies of their peers.

Focus group discussions also revealed that the influence of service quality changes across the life course. It appears to be the most important at the beginning of women's reproductive careers (particularly when they are adolescents). Respondents described how poor experiences

or accounts of poor experiences of young women can, in some cases, mean that a woman is put off using facilities in the future. However, older women who needed long-acting or hormonal methods felt obliged to use the services regardless of perceived facility quality as these methods require engagement with a trained professional.

Despite the ongoing narrative that poor service quality generally discouraged use, there were, nevertheless, a significant number of women (young and older) who said that, even though they may experience substandard care, they would persist with visiting the facilities. In particular, younger respondents, said that even though they knew that they might be poorly received in a public clinic, they would go anyway as, either they had no choice, or they needed the method so badly they were prepared to put up with the negative treatment. Importantly, many young women saw getting a method in the face of potential barriers, as necessary to optimizing their health and wellbeing. In this way, they exhibited agency and self-efficacy to fulfil their goal, despite potentially experiencing negative provider interactions based upon judgment or stigma.

These results are consistent with recent literature that focuses on women's agency to act on their choices and desires, particularly as they relate to meeting their sexual and reproductive health and FP needs [30]. In relation to contraceptive use, agency reflects perceived self-efficacy (a women's belief about her own ability to complete the actions necessary for successful FP use) and behavioral control [31, 32]. Agency can equip individuals with the psycho-social skills to overcome socio-cultural or logistical barriers to accessing health services, including those offering FP [26, 27]. As shown here, the young women (and the women who need long-acting or hormonal methods) find strategies to meet their FP needs, while accepting potential negative experiences at the facility. This suggests that these women value their own needs over the potential treatment they may receive when visiting facilities as a stigmatized group.

Programs are needed to build and reinforce young women (and all women's) agency and self-efficacy to meet their sexual and reproductive health needs; these programs can potentially help to close the equity gap in health service utilization [33]. Women with greater agency may make choices that meet their health needs despite family or societal opposition and regardless of the way they are treated by health care providers [34, 35]. Programs also need to address stigma and discrimination at the facility level that affect individual and community perceptions of service quality. Whole site training and supportive supervision are strategies that have been employed to address these types of barriers to use of services [36]. Once service quality has been improved, it is crucial to spread the word to the broader community and encourage users to share their positive experiences with others to change perceptions of service quality at the community level and reduce this potential barrier to broader service use.

Before concluding, it is important to address some limitations with this study. First, this study that used FGD data was based on women of different demographic backgrounds and user experiences but was not meant to represent any one of these groups. Second, participating women brought their own experiences and those of their friends and families to the discussion but may have also not felt comfortable sharing an experience (positive or negative). Thus, we only have what women were comfortable sharing in a group setting. Finally, the results are not generalizable to urban Burkina Faso nor to Burkina Faso in general since the information only comes from a select sample of women from two cities in Burkina Faso where the Beyond Bias program was implemented in a subset of facilities.

To conclude, improving the quality of FP services based on the locally constructed quality components identified by women should be a specific programmatic goal. These criteria can be identified through quality assessments of facilities with a woman-centered lens adapted to local priorities. In addition, as facility quality is improved, women who visit facilities can be encouraged to share their (ideally) positive experiences with their spouses and networks to

improve community level perspectives of facility quality. Through these multi-pronged actions, women's (and community) expectations and experience of quality can improve. This, in turn, may lead to greater client satisfaction, increased access to women receiving their method of choice, and associated higher contraceptive prevalence as well as less discontinuation. Building women's psychosocial skills through parallel programmatic interventions to improve agency, especially among adolescents and newly married women, may enhance contraceptive uptake and improve their health and well-being and that of their families.

## Acknowledgments

The authors acknowledge those who supported this work at various phases of project design, implementation, and analysis. This includes the data collectors and study participants.

## Author Contributions

**Conceptualization:** Sarah Castle, Fiacre Bazie, Amelia Maytan-Joneydi, Lisa M. Calhoun, Yentema Onadja, Georges Guiella, Ilene S. Speizer.

**Data curation:** Sarah Castle, Fiacre Bazie, Kindo Boukary.

**Formal analysis:** Sarah Castle, Fiacre Bazie, Amelia Maytan-Joneydi, Kindo Boukary, Lisa M. Calhoun, Ilene S. Speizer.

**Funding acquisition:** Ilene S. Speizer.

**Methodology:** Sarah Castle, Fiacre Bazie.

**Project administration:** Amelia Maytan-Joneydi, Kindo Boukary, Yentema Onadja, Georges Guiella.

**Supervision:** Fiacre Bazie, Ilene S. Speizer.

**Writing – original draft:** Sarah Castle.

**Writing – review & editing:** Fiacre Bazie, Amelia Maytan-Joneydi, Kindo Boukary, Lisa M. Calhoun, Yentema Onadja, Georges Guiella, Ilene S. Speizer.

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
