## [Decision Letter · Decision Letter 0]

2 Mar 2023

PGPH-D-22-01766

“You could find a good or a bad provider, I would say you just have to go and see”: A qualitative study of the influence of perceptions of service quality on family planning service use

in Burkina Faso

Dear Dr. Speizer,

Thank you for submitting your manuscript to PLOS Global Public Health. After careful consideration, we feel that it has merit but does not fully meet PLOS Global Public Health’s publication criteria as it currently stands. Therefore, we invite you to submit a revised version of the manuscript that addresses the points raised during the review process.

In your revision, please address the below points suggested by the reviewer:

a- Provide more information on the types of community leaders involved in identifying women to participate in FGDs and how you ensured that this did not lead to biases in the sample or indeed any undue coercion

b- Please address the issue of a  small number of FGDs conducted with men and provide a table and details on this as necessary, and explain whether those were included in the analysis.

c- Please provide context and procedure around the selection of the 2 study urban sites of Bobo & Banfora. Please explain why you state that the results are not generalisable to other cities or urban sites.

Making those minor changes will ensure that the paper is publishable and that  its recommendations can be used by others in similar or adjacent fields of study.

We look forward to receiving your revised manuscript.

Kind regards,

Isabelle Uny

Academic Editor

Journal Requirements:

Additional Editor Comments (if provided):

Reviewers' comments:

Reviewer's Responses to Questions

**Comments to the Author**

1. Does this manuscript meet PLOS Global Public Health’s publication criteria? Is the manuscript technically sound, and do the data support the conclusions? The manuscript must describe methodologically and ethically rigorous research with conclusions that are appropriately drawn based on the data presented.

Reviewer #1: Yes

2. Has the statistical analysis been performed appropriately and rigorously?

Reviewer #1: N/A

3. Have the authors made all data underlying the findings in their manuscript fully available (please refer to the Data Availability Statement at the start of the manuscript PDF file)?

Reviewer #1: Yes

4. Is the manuscript presented in an intelligible fashion and written in standard English?

Reviewer #1: Yes

5. Review Comments to the Author

Reviewer #1: Thank you for the opportunity to review this interesting paper on women's perspectives of seeking family planning from facility providers in Burkina Faso. I have a few small comments that could be considered by the authors:

1) You mention that community leaders were involved in identifying women to participate in FGDs, who were these leaders? Mostly men? Women? Religious leaders? Any chance they could have influenced the sample unduly?

2) There was mention of a small number of FGDs with men, but this is not in the FGD table nor the findings referenced elsewhere that I could see. If these FGDs were not included in the analysis maybe just remove this sentence?

3) A bit of context as to why the two sites of Bobo & Banfora were selected would be helpful. These are both cities, yet you say the results are not generalisable to other cities like Ouaga, why not?

Overall I think this is a solid paper with good recommendations for programmatic change, I recommend to accept with some minor revisions.

6. PLOS authors have the option to publish the peer review history of their article (what does this mean?). If published, this will include your full peer review and any attached files.

**Do you want your identity to be public for this peer review?** For information about this choice, including consent withdrawal, please see our Privacy Policy.

Reviewer #1: **Yes: **Gillian McKay

---

## [Editor Report · Decision Letter 1]

14 Mar 2023

“You could find a good or a bad provider, I would say you just have to go and see”: A qualitative study of the influence of perceptions of service quality on family planning service use

in Burkina Faso

PGPH-D-22-01766R1

Dear Dr. Speizer,

We are pleased to inform you that your manuscript '“You could find a good or a bad provider, I would say you just have to go and see”: A qualitative study of the influence of perceptions of service quality on family planning service use

in Burkina Faso' has been provisionally accepted for publication in PLOS Global Public Health.

Best regards,

Isabelle Uny

Academic Editor